# Platform Design and Preliminary Test Result of an Insect-like Flapping MAV with Direct Motor-Driven Resonant Wings Utilizing Extension Springs

**DOI:** 10.3390/biomimetics8010006

**Published:** 2022-12-23

**Authors:** Seung-hee Jeong, Jeong-hwan Kim, Seung-ik Choi, Jung-keun Park, Tae-sam Kang

**Affiliations:** 1Department of Aerospace Information Engineering, Konkuk University, Seoul 05029, Republic of Korea; 2Department of Smart Vehicle Engineering, Konkuk University, Seoul 05029, Republic of Korea

**Keywords:** flapping-wing micro air vehicle, resonant driving, extension spring, directly driven wings, biomimetic robot, identification

## Abstract

In this paper, we propose a platform for an insect-like flapping winged micro aerial vehicle with a resonant wing-driving system using extension springs (FMAVRES). The resonant wing-driving system is constructed using an extension spring instead of the conventional helical or torsion spring. The extension spring can be mounted more easily, compared with a torsion spring. Furthermore, the proposed resonant driving system has better endurance compared with systems with torsion springs. Using a prototype FMAVRES, it was found that torques generated for roll, pitch, and yaw control are linear to control input signals. Considering transient responses, each torque response as an actuator is modelled as a simple first-order system. Roll, pitch, and yaw control commands affect each other. They should be compensated in a closed loop controller design. Total weight of the prototype FMAVRES is 17.92 g while the lift force of it is 21.3 gf with 80% throttle input. Thus, it is expected that the new platform of FMAVRES could be used effectively to develop simple and robust flapping MAVs.

## 1. Introduction

Wing flapping is the method that all birds and insects use for flying [1]. While a bird uses its tail wing for attitude control, an insect, without a tail wing, uses its flapping wings to control its flight by actively maneuvering its wing flapping motion [2,3,4]. Insects perform better than birds in hovering flight and agile attitude change [4,5]. Thus, insect-mimicking vehicles have the potential for applications in confined small spaces as well as outdoors. However, the design of an insect-mimicking vehicle is technically more challenging than that of bird-inspired tailed vehicle [4].

Several different approaches may be taken to implementing the insect-mimicking wing flapping system. One approach is to use a single DC motor and a complex transmission mechanism to generate flapping motion. Reciprocating flapping motion is obtained by the transmission system using the unidirectional rotary motion of the motor. A pair of wings is tied together via the transmission mechanism so that the wings cannot be controlled individually, and the flapping frequency is the only variable that can be adjusted by the motor. Therefore, servo motors are mounted in this model to produce control force by modifying symmetric wing stroke motion [6,7,8,9,10,11]. Another approach is to use dual unidirectional DC motors. Each motor is equipped with an independent transmission mechanism on each wing [12,13,14,15]. In this model, the frequency of each wing can be controlled individually to produce roll motion. However, two more servo motors are needed for pitching and yawing torque generation via adjusting the wing stroke motion.

Another approach is to use bidirectional actuators to avoid the complexity of the transmission mechanism. In particular, piezo actuators, due to their small size and lightweight, high-power density, and low power consumption, have been chosen for insect-sized flying robots [16,17,18,19,20,21,22]. The long-time research result of the tiny flying robot is the RoboBee, which demonstrated stable controlled flights with the aid of off-board components [17], or uncontrolled takeoff with onboard power source and electronics [19]. However, it still needs more substantial improvement for an efficient stable free flight with onboard electronics.

For larger scale flying robots, such as the hummingbird robot, dual DC motors have been used for directly driving flapping wings instead of the small sized piezo actuators, since the piezo actuators could not supply enough force to lift the larger sized flying robots. The amplitude, position offset, and frequency of each wing can be controlled individually in this platform to generate torques for roll, pitch, and yaw stabilization, as well as lift force for flight. Since there is no transmission mechanism in this system, bidirectional driving of the motor is necessary to generate flapping motions, and the motor should overcome large inertial force, consuming much energy, to reverse the driving direction of flapping motions. To solve this problem, energy storing and reusing via resonance system have been applied.

A resonant structure for dual twin wing, mimicking the muscle and driving structure of insects, is introduced in [23]. Reference [24] shows that the dynamic efficiency can be close to 90% using resonance phenomena. Lau et al. showed that the wing with nonlinear elastic structure consumes only 2% of total mechanical power for flapping at 25 Hz, compared with 23% when there is no elastic storage [25], which confirms the necessity of using elastic springs, notwithstanding the weight increase. Reference [26] suggests the basic platform of a motor-driven flapping-wing system, and demonstrates free flight without a controller. The effect of a resonant flapping system is investigated via simulations and experiments in [27]. Zhang et al. experimentally demonstrated that stabilized hovering flight is possible [28]. Extreme maneuvering of a flapping winged robot with 40 Hz dual motors is demonstrated in [29]. Navigating tight space by sensing driving current for flapping is suggested in [30]. References [31,32] present the design and controlled flight indoors and outdoors of an at-scale dual-motor hummingbird robot. A general framework for understanding the roles of internal damping, aerodynamic and inertial forces, and elastic structures within all spring-wing systems is introduced in [33].

The storing and reusing of inertial energy in flapping is critical to extend the flight time of the robot. For the directly driven dual motor flapping wing system, helical or torsion springs have been chosen for elastic devices [26,27,28,29,30,31,32,33]. In our previous study, we initially applied a torsion spring for a flapping system to implement a resonance flapping wing system [34]. However, we found that aligning the torsion spring axis with the flapping axis is not easy since one side of the spring should be fixed to the center of the rotation axis while the other side should be fixed to the body frame. Also, adjusting the zero position requires precise mounting of the spring to the body frame and rotation axis of the wing. After continuous flapping, the torsion spring often suffered fatigue failure, since flapping required larger displacement than that allowed for the commercially available torsion spring.

In this paper, we introduce a new platform for a motor-directly-driven resonant flapping-wing system using an extension spring (instead of a helical or torsion spring). The basic experimental results of the proposed resonant flapping wing system and discussions are also included. Section 2 explains the structure and natural frequency of the proposed platform together with the mathematical derivation. Section 3 describes the working principles and the experimental results of the prototype flapping winged robot. Section 4 suggests further research topics and provides the conclusions of this paper.

## 2. Flapping Wing System

In this section, we propose the new design of a directly driven flapping wing system with extension spring. The basic working principles are also briefly explained. The dynamics of the flapping system with extension spring and its natural frequency are derived.

### 2.1. Design of the Flapping Wing System with Extension Spring

Figure 1 is the top view schematic of the dual flapping wing-driving system with extension springs. One driving system consists of a DC motor, a pinion gear, a spur gear, a wing, and an extension spring. The motor is fixed to the body frame, and the pinion gear is fixed to the motor to transmit the rotational force of the motor to the spur gear. The wing is fixed to the spur gear. An extension spring is connected between the wing part and the frame. The extension spring can be mounted any place between the body frame and the wing or the spur gear, so long as it tenses the spring, and puts the wing at zero position at rest. The angle between the extension spring and the wing is set to about 150 degrees to have space so that the spring is a little extended at the zero position, which helps the spring work in the same way as a linear spring, as will be explained later.

Figure 1b shows how the driving system works. As the wing starts to rotate in the clockwise direction from the stationary state, that is, the zero point, and performs the down stroke, the extension spring attached to the wing part is stretched by the wing rotation, and elastic energy is accumulated in the extension spring.

When the wing reaches the top of the flap, the rotation direction of the wing is reversed. At this point, the spring is extended to maximum length, and the elastic energy is also maximally stored. When the direction of flapping is reversed and the wing begins to rotate counterclockwise, the elastic energy stored in the spring is released to accelerate the wing to the zero position. As the wing crosses the zero point, a new cycle of energy storing and discharging is repeated.

The frame of the model is manufactured by processing a carbon plate of 1 mm thickness, and to increase the stiffness of the flying robot, a U-shaped carbon plate of 1 mm width is vertically inserted. Two DC motors of 7 mm diameter and 20 mm length are used for driving. The frames of the two motors also work as stiff holding columns for the body of the robot. The weight of a motor is 3.5 gf, and the stall torque is 2.47 N·mm. A metal gear with module 0.3 and 7 teeth is used for the pinion gear, while a plastic gear with module 0.3 and 81 teeth is used for the spur gear. The wings used are the same as those of the KUBeetle [11]. The wingspan is 75 mm, while the wing area is 18.8 cm^2^.

Figure 2 shows the shape of the wings used. The wing membrane is made of thin polyethylene terephthalate, and the thickness is 20 μm. The wing spar is made of 1 mm thin carbon tube.

### 2.2. Dynamics of the Flapping Wing System

The flapping wing system can be modeled as a rotational spring–mass-damper system, as in Equation (1):(1)$$\tau =I_{T}\ddot{\theta }+D_{T}\dot{\theta }+K_{T}\theta$$
where, τ is the torque applied, $I_{T}$ is the moment of inertia of the whole system, $D_{T}$ is the damping coefficient of the total system, and $K_{T}$ is the effective spring coefficient.

The equation of motion according to motor driving is expressed as Equation (2):(2)$$\tau =\left(I_{M}+I_{P}\right)\;{\ddot{\theta }}_{M}+\left(D_{M}+D_{P}\right){\dot{\theta }}_{M}+\tau_{e}$$
where, $\theta_{M}$ is the rotational angle of motor, $I_{M}$ and $I_{P}$ are the moment of inertia of the motor and the pinion gear, respectively, and $D_{M}$ and $D_{P}$ are the damping coefficient of the motor and the pinion gear, respectively. The $\tau_{e}$ is the external torque from the spur gear.

Equation (3) can express the equation of motion according to the rotation of the spur gear:(3)$$\tau_{S}=\left(I_{S}+I_{W}\right)\;{\ddot{\theta }}_{S}+\left(D_{S}+D_{W}\right){\dot{\theta }}_{s}+k_{r}\theta_{S}$$
where, $\theta_{s}$ is the rotational angle of spur gear, $I_{S}$ and $I_{W}$ are the moment of the inertia of the spur gear and the wing, respectively, $D_{S}$ and $D_{W}$ are the linearly approximated damping coefficient of the spur gear and the wing, respectively, and $k_{r}$ is the rotational spring constant of the extension spring.

The relation between the external torque applied by the motor and the reactive torque of the spur gear is expressed as:(4)$$\tau_{e}=\tau_{S}\times \left(\frac{n_{P}}{n_{S}}\right)$$
where, the number of teeth of the pinion gear and that of the spur gear is $n_{P}$ and $n_{S}$, respectively. The relation between the rotational angle of the motor and that of the spur gear is expressed as:(5)$$\theta_{M}=\theta_{S}\times \left(\frac{n_{S}}{n_{P}}\right)$$

Using the abbreviated notation $n=\left(\frac{n_{S}}{n_{P}}\right)$, Equation (1) of the total flapping wing system can be described as:(6)$$\begin{array}{l} \tau =\left(n\left(I_{M}+I_{P}\right)+\frac{1}{n}\left(I_{S}+I_{W}\right)\right){\ddot{\theta }}_{S}\\ +\left(n\left(D_{M}+D_{P}\right)+\frac{1}{n}\left(D_{S}+D_{W}\right)\right){\dot{\theta }}_{s}\\ +\left(\frac{k_{r}}{n}\right)\theta_{S} \end{array}$$

Denoting the total moment of inertia $\left(n\left(I_{M}+I_{P}\right)+\frac{1}{n}\left(I_{S}+I_{W}\right)\right)$ as $I_{T}$, the total damping coefficient $\left(n(D_{M}+D_{P}\right)+\frac{1}{n}\left(D_{S}+D_{W}\right))$ as $D_{T}$, and the effective total spring constant $\left(\frac{k_{r}}{n}\right)$ as $K_{T}$, the natural frequency of the entire system $\omega_{n}$ is expressed as $\sqrt{\frac{K_{T}}{I_{T}}}$, and the damped natural frequency $\omega_{d}$ is given as $\omega_{n}\sqrt{1-\zeta^{2}}=\sqrt{\frac{K_{T}}{I_{T}}}\times \sqrt{1-\zeta^{2}}$.

The rotational elastic constant can be derived from the torque generated from the extension spring. Figure 3 shows the motion of extension spring when the spur gear and wing are rotated from (a) to (b). Referring to Figure 4, when $d_{1},\;d_{2},\;r$, the rotation angle $\theta_{s}$ of the spur gear, and the rotation angle α of the extension spring are given, the torque by the extended spring is obtained as:(7)$$\tau_{es}=k_{e}\left(d_{2}-d_{1}\right)\times \cos\left(\frac{\pi }{2}-\left(\theta_{S}+\alpha \right)\right)\times r$$
where, $k_{e}$ is the linear elastic constant of the extension spring, $d_{1}$ is the spring length at zero position without extension, $d_{2}$ is the spring length when it is extended according to the rotation of the spur gear, and r is the distance between the center of the spur gear, and the point where the spring is fixed on the spur gear. The $k_{e},\;d_{1}$, and r are fixed constants, and when $\theta_{s}$ is given, $d_{2}$ and α are determined. The angle of the spur gear and the rotation angle of the wing are the same. Thus, the torque in Equation (7) is the function of the wing rotation angle $\theta_{W}$.

The torque enforced by the extension spring to the spur gear with the wing rotation angle $\theta_{W}$ is shown as the blue line in Figure 5. Here the $k_{e}$ is 0.796 N/mm, the $d_{1}$ is 13.4 mm and the r is 5 mm.

The slope of the graph in Figure 5 is the rotational elastic constant $k_{r}$. When the wing rotation angle is small, the torque generated by the extension spring to the spur gear is minor.

By stretching the extension spring at the zero position by moving away the fixing point to the body frame, tension can be provided, even when the wing is at zero position. In this case, the spring torque at low wing angle range is increased, compared with the toque increase in higher angle range, and thus the rotational elastic constant is linearized. Denoting the increased distance offset as $\bar{d}$, Equation (7) is modified into Equation (8). The red line in Figure 5 shows that when the offset distance $\bar{d}$ is given, the effective torque with wing rotation becomes more linear. Here, the $\bar{d}$ is fixed as 2.4 mm.
(8)$$\tau_{es}=k_{e}\left(d_{2}+\bar{d}-d_{1}\right)\times \cos\left(\frac{\pi }{2}-\left(\theta_{s}+\alpha \right)\right)\times r$$

With the given $\bar{d}$ and using root mean square method, the rotational elastic constant $k_{r}$ is obtained as 0.01954 N·mm/rad, and the correlation factor is 0.9965. The number of teeth of pinion gear and spur gear are 7 and 81, respectively. The moment of inertia of all of the rotating components, which is computed by CATIA simulation program, is 7.53 × 10^−8^ g·m^2^. Therefore, the natural frequency of the whole flapping system is around 22 Hz, as calculated from Equation (6) with the given parameters $k_{e}$ and n.

## 3. Prototype Flapping Winged Robot

Figure 6 shows the prototype flapping winged robot with extension springs. The body frame coordinate is also defined in Figure 6. The roll, pitch, and yaw control torque can be generated by adjusting the amplitude, offset, and frequency of the control input signal, respectively.

Table 1 shows weight distribution. The total weight is estimated to be 17.92 g. Weights of the two motors account for 37.8% of the total weight. The weight of the spur gear can be reduced if the unused part is cut out. Weights of the control board and dual motor driver are estimated ones. The dominant weight component in the dual motor driver is the capacitor used for stabilizing the input voltage.

### 3.1. Control Torque Generation of the Flapping Winged Robot

Figure 6 shows the coordinate system of the robot. The front side of the model is the negative X axis direction. The positive Y axis is located at the left side of the X axis. The positive Z axis is in the opposite direction to the gravity force.

Figure 7 shows the wing flapping motion for the torque control of each axis [31]. The yellow shaded section indicates the sweeping area of wing for each motion. Wider area produces more lift force, and asymmetric lift force invokes torque for control. Therefore, the roll, pitch, and yaw motions can be controlled by varying the wing flapping characteristics. These can be realized by adjusting the amplitude, offset, and frequency, respectively, of the sinusoidal signal that drives the flapping. By varying the amplitude of the flapping on the left and right wings, the torque for roll control is invoked. By giving offset value to the motor driving sinusoidal signal, the torque for pitch control is generated. By split signal driving (i.e., giving variation to the up stroke and down stroke speed, as shown in the right side of Figure 7, the torque for yaw control can be obtained). Figure 8 shows how the sinusoidal driving signal is modified for roll, pitch, and yaw torque generation. For all control commands, the voltages are normalized. In Figure 8, in upstroke motion, using PWM, the driving voltage is being increased, while in downstroke motion, the driving voltage is being decreased.

To control the wing flapping system, the motor drive signal should be similar to the sinusoidal wave of 22 Hz, which is the natural frequency. To generate the driving and control signal, an Mbed LPC-1768 microcontroller board is used, which is equipped with ARM Coretex M3. Two SZH-MDBL-010 (SMG) boards are used to drive the two motors. The driver board uses MX1616H, which can be operated with voltage supply range of (2 to 10) V, and the peak driving current is 2.5 A. The single motor driver receives the PWM signal from the micro controller, and drives one motor. The single motor driver has two PWM outputs, i.e., (+)PWM output and (−)PWM output. One terminal of the motor is connected directly to the (+)PWM output, while the other terminal is connected directly to the (−)PWM output. An external 7.4 V power is used for the driving. The voltage that can be obtained from two serially-connected lithium–ion batteries is 7.4 V. The microcontroller generates, for one motor, 2 PWM outputs and two control signals for sign control, which are used to generate the roll, pitch, and yaw control torques, as well as lift force. The sinusoidal driving signal is obtained by adjusting the PWM duty rates. To drive the (+) signal, the (+)PWM output is actively driven, while the (−)PWM output is kept to be ground; to drive the (−) signal, the (+)PWM output is kept to be ground, while the (−)PWM output is actively driven. The custom small printed circuit control and driving board will be fabricated and mounted to the flying robot soon.

### 3.2. Experimental Results for Control Torque Generation

Lift force and torques for roll, pitch, and yaw control torques generated by applying the modulated signals were measured using a load cell, Nano17 IP68 SI-50-0.5 6 axis F/T sensor from ATI. The bottom frame of the flapping winged robot was bolted to the load cell, as shown in Figure 9. It was tested inside a arcrylic box that blocks the external factors affecting the test results.

Table 2 shows the control inputs of throttle, roll, pitch, and yaw commands. To get the best energy efficiency, the nominal flapping frequency is set as 22 Hz, which is the natural frequency of the system. The nominal throttles for all inputs are set at 80%, so that the control signals can use the remaining 20% throttle. For the thrust test, 60%, 80%, and 100% throttles are applied.

The Nano17 load cell measures 1600 samples per second, and records data for 10 s. The torques (N·mm) in the roll, pitch, and yaw directions, as well as the forces (gf) along the x, y, and z axes are measured.

In the first experiment, the responses to the positive and negative inputs of roll, pitch, and yaw were measured for 4 s each. The average of each signal was obtained after the high-frequency noises were filtered out using a low pass filter with 30 Hz bandwidth. At the second experiment, we measured the step responses to the control inputs in order to see the transient responses.

Figure 10 shows the results of the first experiment. The torques for roll, pitch, and yaw are approximately linear to each command input. The cross coupling is small, but not negligible. Figure 10d shows that the lift forces are 15.5 gf, 21.3 gf, and 24.1 gf at 60%, 80%, and 100% thrust input, respectively. It is expected that the flapping winged robot will have enough thrust for flight if the weight of the flying robot is kept at less than 20 g.

In the second experiment, transient responses to step inputs for the roll, pitch, and yaw commands are measured. After applying 80% throttle commands for 2 s, the step commands are applied for 2 s. The magnitudes of the roll, pitch, and yaw step command inputs are +10% amplitude increase, +20% offset increase, and +8 Hz difference in split cycles, respectively.

Figure 11 shows the step responses to roll, pitch, and yaw command inputs. The black lines represent the measured outputs (which are averaged). Even though they fluctuate a lot due to flapping vibrations, the outputs can be modeled as first-order system responses. Coupling effects of roll, pitch, and yaw commands are not negligible. In particular, the roll command is highly coupled to the yaw command. About 0.44 N·mm roll toque is generated by the yaw command to generate 0.72 N·mm yaw torque.

The Matlab System Identification Toolbox was utilized to obtain the first-order transfer functions between the inputs and outputs. The outputs of the identified models are shown as colored lines in the graphs in Figure 11. The magnitudes of the input signals were rescaled as 1.5 for identification. Table 3 shows the obtained first-order transfer functions for each measured data. The experimental data and simulation outputs match well (except for high-frequency noise). Furthermore, the proposed flapping system with an extension spring shows good endurance. The flapping system with an extension spring has been working for more than an accumulated 1 h without fatigue failure since it was fabricated, while the one with torsional spring broke before 1 min of continuous operation [35]. This is due to the fact that by adjusting the r in Figure 3, the flapping system with an extension spring can be designed that is always working in the allowed operating range, while the flapping system with torsion spring requires larger displacement than is allowed for commercially available torsion springs. The displacement amplitude of the torsional spring is exactly the same as the wing flapping amplitude and cannot be adjusted.

## 4. Conclusions

In this paper, the platform of a directly motor driven resonant flapping MAV using an extension spring is devised. Through theoretical and experimental analysis, it is found that the proposed flapping system has better endurance than the conventional torsion spring, demonstrating more than one hour accumulated operating time since it was fabricated. Mounting the extension spring is easy, and the mounting position can be adjusted freely (so long as it does not block the flapping motion).

With the fabricated prototype flapping robot, the lift force is 21.3 gf with 80% thrust input. Generated control torques for roll, pitch, and yaw, which can be modelled as simple first-order systems, are coupled to each other strongly. In particular, the yaw command to generate 0.72 N·mm yaw torque invoked 0.44 N·mm roll torque.

The total weight of the prototype, including estimated weight of electronic boards, is 17.92 g which is small enough for flight considering a lift force of 21.3 gf with 80% throttle input. Thus, it is expected that the proposed platform could be utilized effectively to develop long endurance directly motor driven resonant flapping MAVs. Further study should be conducted for real flight after developing and mounting onboard sensors, a control board, and a battery power source for the flying robot.

## Figures and Tables

**Figure 1 biomimetics-08-00006-f001:**
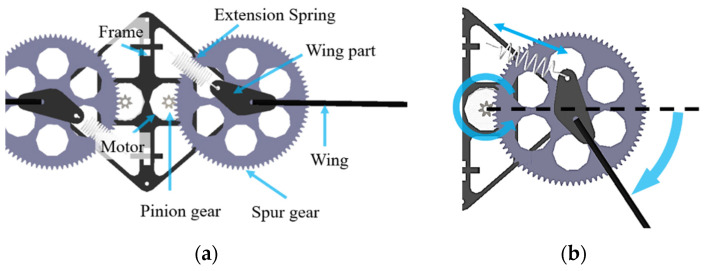
Top view of the flapping wing systems: (**a**) wing at rest; (**b**) wing rotated.

**Figure 2 biomimetics-08-00006-f002:**
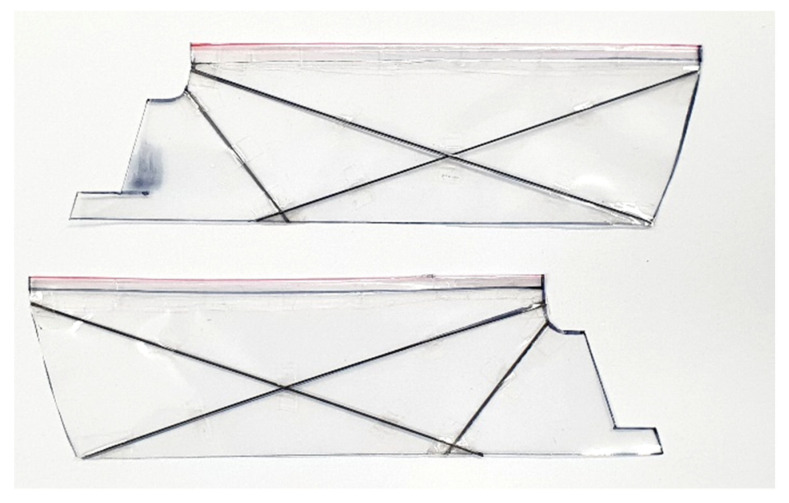
Picture of the wings used.

**Figure 3 biomimetics-08-00006-f003:**
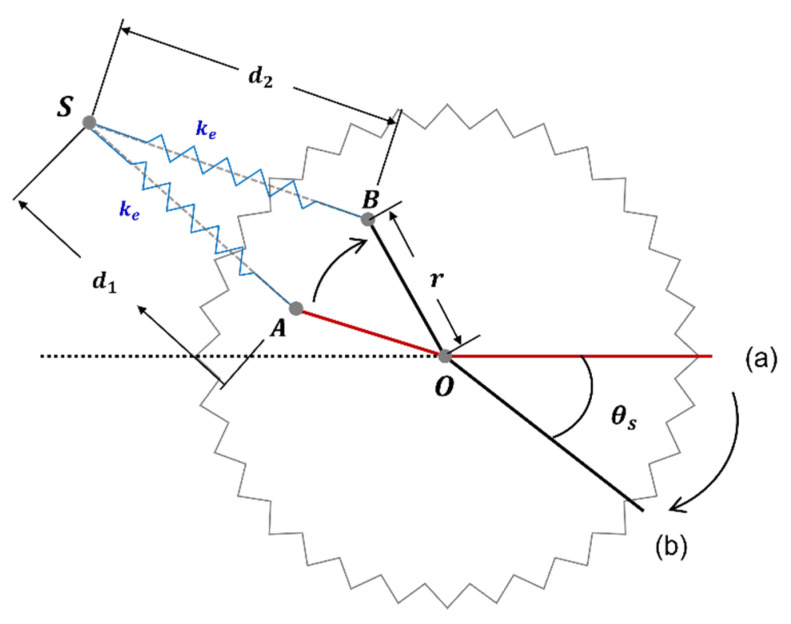
The motion of the extension spring with the spur gear movement.

**Figure 4 biomimetics-08-00006-f004:**
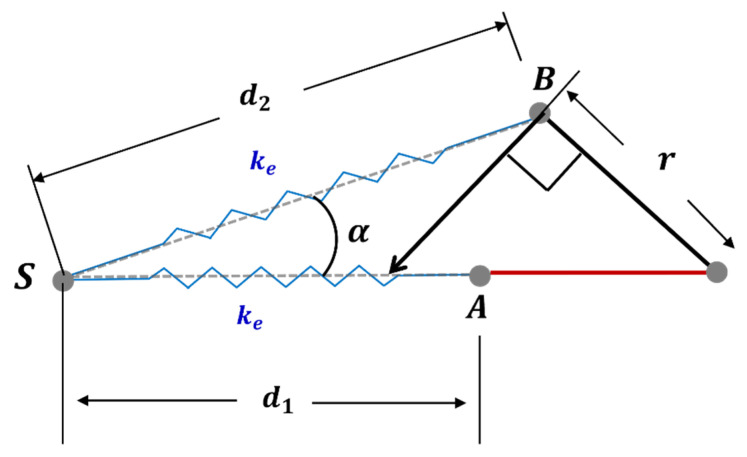
Torque acting on the spur gear by the extension spring.

**Figure 5 biomimetics-08-00006-f005:**
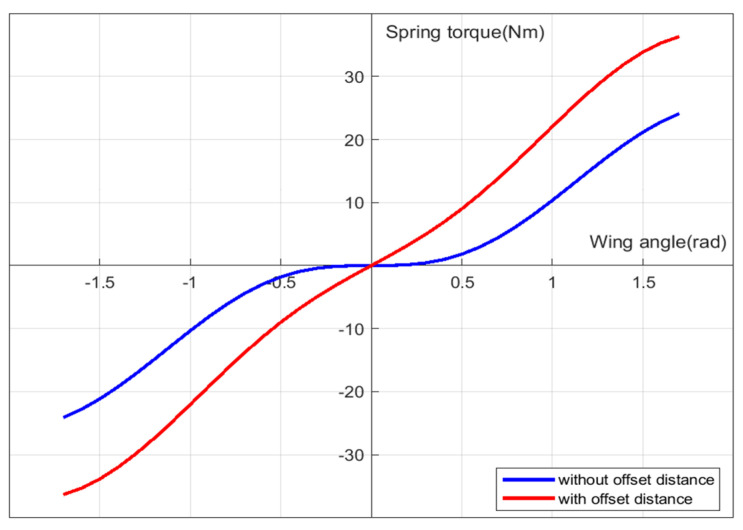
Spring torque with wing rotation angle.

**Figure 6 biomimetics-08-00006-f006:**
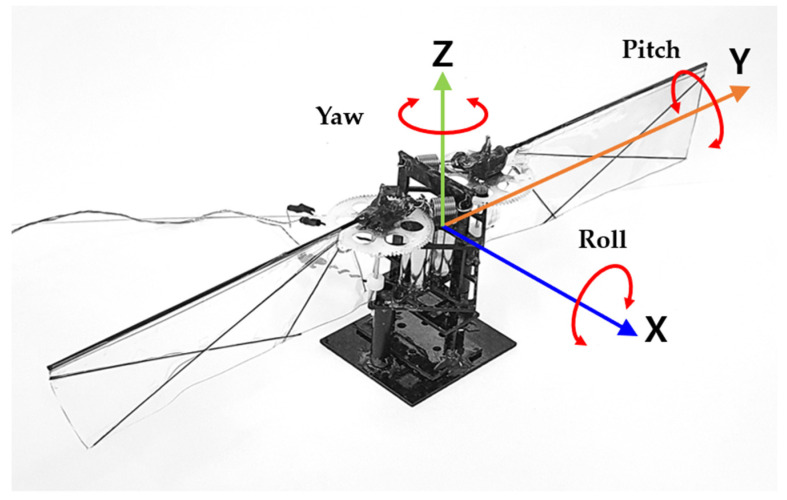
Prototype of the flapping winged flying robot with extension springs and the body frame coordinate definition.

**Figure 7 biomimetics-08-00006-f007:**
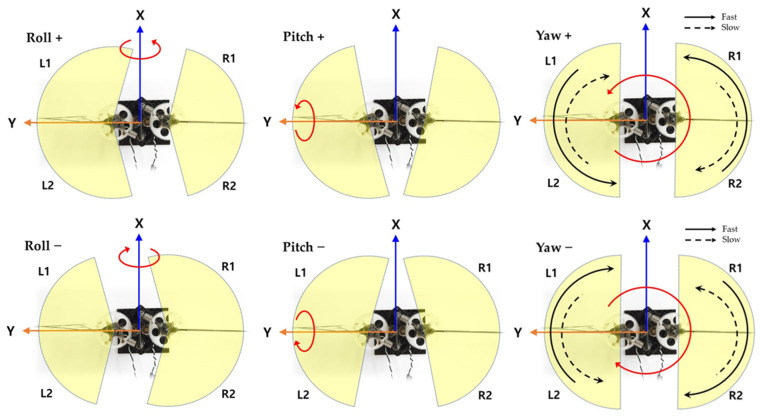
Wing flapping variations for roll, pitch, and yaw maneuvering.

**Figure 8 biomimetics-08-00006-f008:**
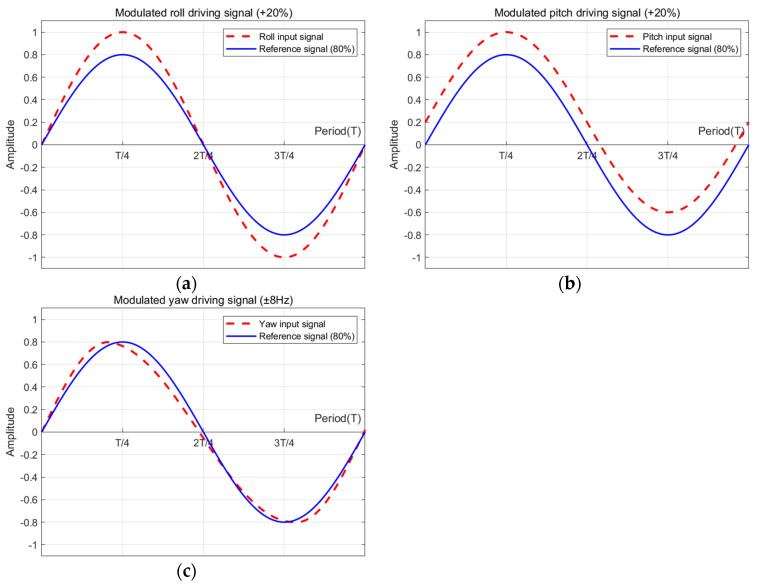
Driving signal waveforms for attitude controls: (**a**) roll control; (**b**) pitch control; (**c**) yaw control.

**Figure 9 biomimetics-08-00006-f009:**
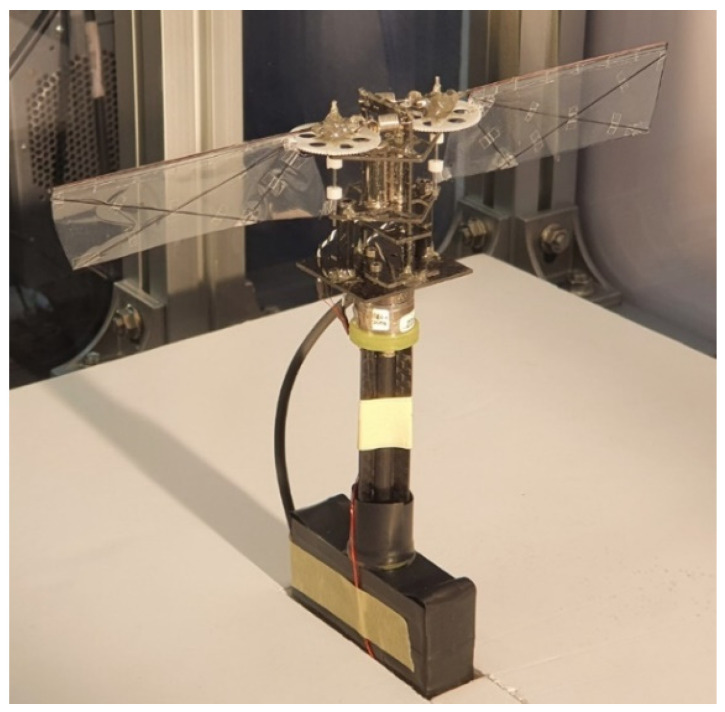
Experimental setup for measuring torques and lift forces with roll, pitch, and yaw control inputs.

**Figure 10 biomimetics-08-00006-f010:**
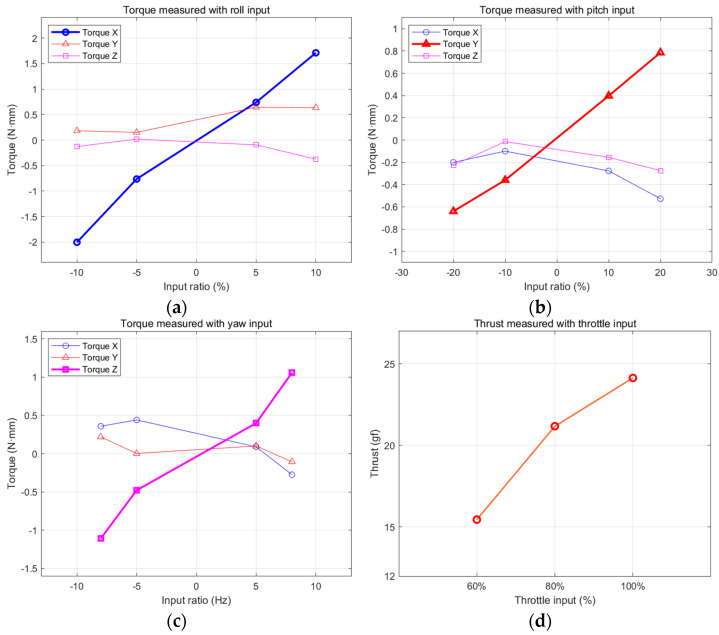
Steady state torques and lift force: (**a**) torques with roll command input; (**b**) torques with pitch command input; (**c**) torques with yaw command input; (**d**) lift force with 60%, 80%, and 100% thrust input.

**Figure 11 biomimetics-08-00006-f011:**
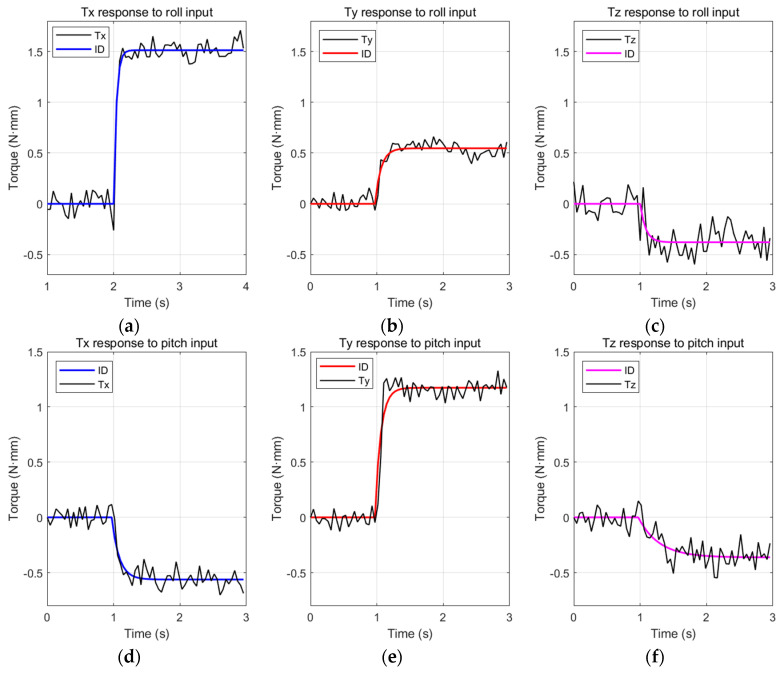
Step responses: (**a**) torque x response to +10% roll input; (**b**) torque y response to +10% roll input; (**c**) torque z response to +10% roll input; (**d**) torque x response to +20% pitch input; (**e**) torque y response to +20% pitch input; (**f**) torque z response to +20% pitch input; (**g**) torque x response to +8 Hz split cycle yaw input; (**h**) torque y response to +8 Hz split cycle yaw input; (**i**) torque z response to +8 Hz split cycle yaw input.

**Table 1 biomimetics-08-00006-t001:** Weight of each component.

Component	Unit Weight (g)	Quantity	Sum (g)
DC Motor	3.50	2	7.00
Pinion gear	0.05	2	0.10
Spur gear	0.37	2	0.74
Wing	0.21	2	0.42
Wing shaft	0.59	2	1.18
Frame	2.32	1	2.32
Extension spring	0.22	2	0.44
Adhesive and wire	0.82	1	0.82
Control board *	1.50	1	1.50
Dual motor driver *	1.20	1	1.20
Battery	1.10	2	2.20
Total weight (g)			17.92

* Estimated.

**Table 2 biomimetics-08-00006-t002:** Test control inputs for throttle, roll, pitch, and yaw responses.

	Throttle (%)	Offset (%)	Frequency (Hz)
Thrust	(80 ± 20)	0	22
Roll	(80 ± 10), (80 ± 5)	0	22
Pitch	80	±20, ±10	22
Yaw	80	0	(22 ± 8), (22 ± 5)

**Table 3 biomimetics-08-00006-t003:** Transfer function for each normalized input.

	Torque (Tx)	Torque (Ty)	Torque (Tz)
Roll input (+10%)	$\frac{24.44}{s+24.24}$	$\frac{4.068}{s+11.15}$	$\frac{-1.899}{s+7.556}$
Pitch input (+20%)	$\frac{-3.355}{s+8.975}$	$\frac{\;9.235}{s+11.8}$	$\frac{-0.766}{s+3.189}$
Yaw input (+8 Hz)	$\frac{5.227}{s+17.46}$	$\frac{-1.628}{s+11.22}$	$\frac{5.415}{s+11.26}$

## Data Availability

Not applicable.

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
