# Peer review of "Platform Design and Preliminary Test Result of an Insect-like Flapping MAV with Direct Motor-Driven Resonant Wings Utilizing Extension Springs"

_biomimetics, 2022, doi:10.3390/biomimetics8010006_

Round 1

Reviewer 1 Report

1.       The abstract needs to be modified. Please read more abstracts of related articles and rewrite it effectively

2.       Why do you introduce springs? It causes more weight burden to these flapping wing vehicles? It will not be efficient choice when it is flown

3.       Is it natural frequency or flapping frequency. How do you arrive it as 22 Hz?

4.       How do you say it is 20gf force is increased? What was the baseline?

5.       Your driving signal is sine wave as shown in Fig. 8 and your output lift force shown in Fig. 10d is having constant value for 1 to 5 sec.  Please clarify

6.       Where is the thrust component? It is not plotted

7.       You have to make lot of measurements and understand the lift and thrust characteristics. By changing the flapping frequency, wind speed, angle of attack and other parameters which affects the lift

8.       Fig. 11.  Is it step response?

9.       How are the transfer functions are arrived. Please explain

10.   The conclusion is poorly written. What are your major inferences and scientific interpretations?

Reviewer 2 Report

The paper presents a prototype of a flapping winged micro aerial vehicle with the focus on the hardware structure and the flapping wing dynamics modeling. Overall the paper is well structured, but due to the declared stage of development, limited results are presented, and future development are needed.

If there is ready the control algorithm its description can be inserted in the paper.

The presentation of the state-of-the art section can be improved by revising the language.

Other specific points that are recommended to be revised are listed below:

lines 14-15: "Extension spring is used because attaching the extension spring to the resonant driving system is easier compared with the case using torsion spring."

line 78: " ... found that align the torsion spring ...";

line 19: "more than 20g" better to mention upper limit eventually with a tolerance.

line 123: "The weight of a motor is 3.5 gf and the stall toque is 2.47 ???"

line 138,140: "rotation of the motor" ? or its shaft?

The place of the caption for Fig. 9 should be checked;

line 250-251: unfinished statement "In the first experiment, responses to the positive and negative inputs of roll, pitch, and yaw for 4 seconds each."

Round 2

Reviewer 1 Report

Please provide the weight of each components of Flapping wing system as a Table

Try to improve the conclusion with salient points of inference instead of more general terms

Still i am wondering about real time flight tests. Have you  ever tried with the torsion springs.  If so Please discuss in the manuscript

Still the sentence formation is not good in the abstract and conclusion. Please ask native english speaker or writer to modify it
